# Recent Progress on Vaccines Produced in Transgenic Plants

**DOI:** 10.3390/vaccines10111861

**Published:** 2022-11-03

**Authors:** Goabaone Gaobotse, Srividhya Venkataraman, Kamogelo M. Mmereke, Khaled Moustafa, Kathleen Hefferon, Abdullah Makhzoum

**Affiliations:** 1Department of Biological Sciences & Biotechnology, Botswana International University of Science & Technology, Palapye, Botswana; 2Virology Laboratory, Department of Cell & Systems Biology, University of Toronto, Toronto, ON M5S 3B2, Canada; 3The Arabic Preprint Server/Arabic Science Archive (ArabiXiv);; 4Department of Microbiology, Cornell University, Ithaca, NY 14850, USA

**Keywords:** vaccine, transgenic plant, plant-made vaccine, plant molecular pharming, recombinant protein, expression, downstream processing, immunogenicity

## Abstract

The development of vaccines from plants has been going on for over two decades now. Vaccine production in plants requires time and a lot of effort. Despite global efforts in plant-made vaccine development, there are still challenges that hinder the realization of the final objective of manufacturing approved and safe products. Despite delays in the commercialization of plant-made vaccines, there are some human vaccines that are in clinical trials. The novel coronavirus (SARS-CoV-2) and its resultant disease, coronavirus disease 2019 (COVID-19), have reminded the global scientific community of the importance of vaccines. Plant-made vaccines could not be more important in tackling such unexpected pandemics as COVID-19. In this review, we explore current progress in the development of vaccines manufactured in transgenic plants for different human diseases over the past 5 years. However, we first explore the different host species and plant expression systems during recombinant protein production, including their shortcomings and benefits. Lastly, we address the optimization of existing plant-dependent vaccine production protocols that are aimed at improving the recovery and purification of these recombinant proteins.

## 1. Introduction

The last two decades have seen an escalation in the importance of plant-made vaccines. This is chiefly because conventional vaccine production strategies are often surrounded by issues of safety and inadequate efficacy. Plant-made vaccines can circumvent these shortcomings and allow the production of recombinant proteins in large numbers. Additionally, the use of transgenic plants in the production of vaccines eliminates issues of contamination from mammalian-borne pathogens. The production of vaccines using plant cells as production hosts is also cost-effective and affords the production of complex antigens [1]. Recombinant protein production involves the use of different host expression systems. Each of these systems has its own advantages and disadvantages that pertain to, but are not limited to, their cost of operation, length of time of production, yield, and many other factors.

The COVID-19 pandemic has emphasized the importance of vaccines, especially against viral diseases. The emergence of diseases such as COVID-19 requires solid solutions for their control and prevention. There is an increase in the demand for quality recombinant vaccines, which increases the need for new technologies for recombinant vaccine production. This increasing demand for recombinant proteins has afforded a platform for recombinant protein production using different expression hosts. Therefore, plant molecular pharming (PMP) has continued to evolve in the advancement of transient protein expression. This has, in turn, led to increased protein yield while at the same time reducing the time it takes to produce these proteins. Currently, mammalian cell cultures, insect cells, bacteria, and yeast expression host systems remain in use mainly because they are well-defined [2]. Even though yeast expression systems offer post-translational modifications and bacterial expression systems allow for the high yield of recombinant proteins in a short period of time, most of the approved biopharmaceuticals are products of mammalian cell culture expression systems and viral vectors [3,4].

As plant molecular pharming continues to advance and recombinant protein production processes continue to improve, here we review the different host plant species that are in common use for recombinant protein production. We also assess and explore the plant recombinant protein expression system and other expression systems that are in use. This review also assesses progress made over the past 5 years in the production of vaccines for different significant and terminal human diseases such as cancer and HIV [5]. Lastly, we consider ways through which recombinant protein production and biosynthesis can be optimized and review downstream processing and purification techniques during plant-based vaccine production.

## 2. Host Plant Species Used during Recombinant Protein Production

Plant molecular pharming (PMP) has afforded the production of pharmaceuticals such as vaccines in plants. Transgenic plants allow the production of recombinant proteins that are essential for disease prevention, diagnosis, and treatment. The choice of a host plant species during the production of vaccines is very important.

Tobacco is a host plant that can be used in the production of different therapeutic proteins such as cytokines, vaccines, and antibodies [6]. The use of tobacco is simple and convenient, and, additionally, tobacco provides abundant material for the characterization of proteins [7]. Tobacco also has a high soluble protein content [6]. Transgenic tobacco leads to a high expression of transgenes [8]. *Arabidopsis thaliana* as a transgenic plant host species during recombinant protein production allows easy *Agrobacterium*-mediated transformation [9] by the floral dip method. Additionally, this plant possesses small genetically tractable genomes that allow ease of transformation by genetic engineering techniques while producing many seeds. Therefore, it has been regarded as a model organism. However, *A. thaliana* is not suitable for large-scale production because of its small foliage size, which leads to low biomass yield. Additionally, it cannot be cultivated at a large scale, making its use in large-scale molecular pharming very problematic [9]. Rice is one of the four cereals (along with wheat, maize, and barley) that are commonly used in recombinant protein production. In comparison to maize, the annual grain yield of rice of 6600 kg/ha is slightly lower, and rice seeds also possess a slightly lower protein content (about 8%) than maize seeds [9]. Nonetheless, rice remains a model cereal species as its complete genome has been sequenced [10,11]. Perhaps the biggest shortcoming of rice as a host plant organism, when compared to maize, is that it is expensive to purchase. This may deter some countries from using it during recombinant protein production. However, according to [9], this may not necessarily be the case in Africa or Asia because, in these continents, rice is commonly grown as a staple food. Among the advantages of maize is that it has the highest grain yield compared to other cereals, with an annual yield of approximately 8300 Kg/ha. Maize also has a very high seed protein content of about 10%. Additionally, it is the most widely cultivated crop plant in North America [9]. Because of this reason, it is relatively cheaper to purchase when compared to other cereals such as rice. Maize can be transformed very easily in vitro and can be produced very quickly in the ploughing fields. The company ProdiGene Inc. has two commercial products that are expressed in maize [12,13]. Among the tube plants, the potato has a very high annual tuber biomass yield of about 125 tonnes/ha. For this reason, potatoes can be employed during the production of large quantities of recombinant proteins because potato tubers are adapted to accommodate large quantities of proteins [9]. The use of potatoes for vaccine delivery is hindered by the fact that potatoes are cooked before being consumed, and heat can denature recombinant proteins and inhibit their capacity to induce an immune response. Additionally, potatoes contain a lot of starch that may interfere with the downstream processing of pharmaceutical proteins. Tomatoes are host plants that can be cultivated in greenhouses, an advantage that can allow their mass production and containment. Tomato plants have a high annual fruit biomass yield of around 60 tonnes/ha and have been used mainly in the production of vaccine candidates [14]. Algae as a recombinant protein production host can be generated very rapidly and, therefore, can be scaled up in a short period of time, something that makes it ideal for recombinant protein production. Additionally, as demonstrated by [15], the use of algae in recombinant protein production is cost-effective because the media used to culture algae is not expensive and it can be reused to culture algae grown in a continuous cycle. Lettuce grows very rapidly and yet produces very low levels of secondary metabolites such as phenolics and alkaloids [16]. This makes it ideal for recombinant protein production as the low levels of secondary metabolites simplify the protein purification process and reduce the cost of production. Although it has an annual biomass of 30 tonnes/ha, lettuce is quite expensive to produce and, subsequently, to purchase [9]. Additionally, lettuce contains a lot of water (98%), which impacts negatively on protein stability and yield. Although lettuce is important in the production of human vaccines, the stability of proteins in harvested materials is often low [9].

Table 1 summarizes some of the applications of these plant host species that have been reported over time.

## 3. Different Expression Systems Used during Recombinant Protein Production

The development of recombinant protein technology in the 1970s and the discovery that eukaryotic DNA can be expressed in *E. coli* [37] were very significant moments in the progression of molecular biology. For a long time, mammalian cells and bacterial cells, especially *E. coli,* were the commonly used expression systems for recombinant protein production. However, due to their cost effectiveness and ease of cultivation, plants have become pervasive in recombinant protein expression, especially in the development of vaccine candidates for different diseases. Some of these vaccines have undergone clinical trials [38,39,40,41,42,43]. The selection of a recombinant protein expression system is a very important step. Plants, mammalian cells, bacteria, yeasts, and insect cells are the most common expression systems in use. The commonly used mammalian cells in recombinant protein production are animal cells, while the commonly used bacterium is *E. coli* [44]. *Pichia pastoris* and *Saccharomyces cerevisiae* are the most used yeasts in recombinant protein technology [45].

### 3.1. Plants

Protein-based biologics are one of the fastest-growing and largest group of pharmaceutical products [46]. Transgenic plants provide a cost-effective, easily scalable, and safe platform for recombinant protein production. The successful application of the anti-Ebola virus drug ZMapp™ [47] is one example of the remarkable progress of PMP during recombinant protein production. Minor differences between plant and human cells with respect to their N-glycosylation patterns have been an important issue, as they may elicit the expression of plant-glycan-specific antibodies that could decrease the therapeutic efficiency of the plant-produced protein and likely result in adverse effects. Plant glycoengineering knocks out specific genes necessary for plant-specific glycosylation patterns while incorporating mammalian glycosylation genes, leading to the generation of plant hosts that can express mAbs containing genuine human N-glycans. Moreover, these plant-expressed mAbs display a degree of glycan homogeneity that cannot be generated by mammalian cells or in vitro expression systems. This provides an advantage at the stage of regulatory approval of the concerned plant-based protein/therapeutic. Therefore, the availability of plant lines that can express biologics containing tailor-made mammalian N-glycans based on demand affords the development of therapeutics and vaccines with more robust efficacy and safety than those generated using other production platforms [46]. Plants eliminate the problem of contamination, which is common with the use of other expression systems such as bacteria [46]. Although the bacterial species *Agrobacterium* has been known as a plant pathogen since the early 20th century, its ability to introduce foreign DNA has only been exploited for the past few decades [48]. Agrobacterium tumefaciens is the causative agent of crown gall disease in a wide range of host plant species wherein it transfers and integrates a part of its own DNA, called T-DNA, into the plant genome [49]. Agrobacterium enters the plant through wounds or cuts occurring in its root, stem, or leaves following which it inserts its plasmid T-DNA and triggers the plant to develop swollen galls. The bacterium is able to perform interkingdom DNA transfer, which makes it a promising vector for producing foreign proteins in transgenic plants. For the purpose of genetic engineering, the virulence genes in Agrobacterium have to be removed, or, in other words, the bacterium has to be “disarmed” by the elimination of most of the T-DNA except for the right and left border sequences, by means of which the foreign gene is integrated into the genome of plant cells. Figure 1 below shows the different steps for bioengineering vaccines and express them in transgenic plants.

Despite the promise of plant-based protein production systems, there are issues of regulatory approval that have to be met for the appropriate control of large-scale and commercial production of recombinant pharmaceuticals and biologics in plants. Because the production of recombinant proteins in cell cultures such as microbial or mammalian cells is capital and labor-intensive, the pervasiveness of plant-based recombinant protein production platforms is a welcome development. Unlike cell culture-based systems, plant production systems afford increased safety and the upstream expression of recombinant proteins.

### 3.2. Mammalian Cells

The use of mammalian cells requires careful consideration of the cell line to use. A decision to use a specific mammalian cell line is governed by factors such as the required production scale of the recombinant protein, post-translational modifications, and whether protein expression is stable or transient [50]. Chinese hamster ovary cells (CHO) and human embryonic kidney 293 cells (HEK-293) are the most commonly used mammalian cells during recombinant protein production [51] because these cells can be utilized both as adherent cultures or in suspension. Mammalian cells are the only platform that allows the production of recombinant proteins that are as genetically similar as possible to the wild type. This is truer for antibodies produced in mammalian cells that are almost the same as human-produced antibodies [52]. Because of their ability to perform complex post-translational modifications, mammalian cells are commonly used for the production of complex recombinant proteins [51]. The downside to the use of mammalian cells as a recombinant protein production platform is that culturing mammalian cells requires specialized facilities and expensive culture media [46].

### 3.3. Insect Cells

Insect cells are also better utilized in the production of proteins with complex post-translational modifications. Unlike mammalian cells, insect cells are much easier to culture and maintain in vitro and can achieve both transient and stable protein expression [53]. Insect cells also have higher tolerance to osmolality and higher protein expression levels [54]. *Drosophila melanogaster* is amongst the most used insect cell lines during recombinant protein production. As is the case with mammalian cells, the choice of an insect cell line to use during recombinant protein production depends on the method of protein expression, whether it is transient or stable [55]. The use of insect cells in conjunction with a baculovirus expression vector system (BEVS) is becoming very pervasive as a recombinant protein production platform [54]. There is active research on the elimination of serum from insect cell culture media, which would be greatly beneficial in eliminating xenogeneic contamination.

### 3.4. Bacteria

Bacteria are quite easy to manipulate for genetic engineering. Originating in the gut, *E. coli*, due to its in vitro malleability, phenotypic, and genetic diversity, is the most widely utilized bacterial host organism during recombinant protein production. It is amongst the most diverse bacteria with very little conservation among its different strains [56]. There has been active research aimed at improving the capability of *E*. *coli* as a host species during recombinant protein expression. This includes deleting some genes from the *E. coli* genome in order to get rid of damaged genes, insertion sequences, and transposons [57]. Regulatory approval for the use of bacteria is in place, and these microbes do not require a long period of time to grow.

### 3.5. Yeasts

Yeasts can be easily manipulated genetically, making them ideal for recombinant protein production. Although *Saccharomyces cerevisiae* remains the widely used yeast for recombinant protein production, there has been an emergence of other species of yeasts such as *Yarrowia lipolytica, Komagataella* sp., and *Kluyveromyces lactis* that have shown potential in recombinant protein expression [45]. Perhaps the most desirable attribute of yeasts is their ability to make post-translational modifications post-recombinant protein development, which increases the stability of the produced proteins [58]. Additionally, because yeasts are tolerant to low pH, they can be applied during the large-scale production of recombinant proteins. Table 2 summarises the advantages and disadvantages of different protein expression platforms.

There is no doubt that plants are gaining traction as an ideal system to produce recombinant proteins. However, the commercialization of plant systems in this regard is not forthcoming. Although downstream processing and inconsistent product quality may be to blame, relatively low yields of recombinant proteins produced in plants have also been pointed out. Recombinant protein yield is indicative of the intrinsic productivity of the host plant. The three main targets of a recombinant protein expression system are low cost, high quality, and high yield [2]. The scalability and low cost of plant cultivation were expected to reduce the manufacturing costs of plant-based expression systems. However, this has not been the case, due in part to the low yields of plant systems as well as the product recovery and purification costs. While cell-specific productivity (qP), which indicates the daily production capacity of a particular system, has been reported for other systems such as CHO [62], it is extremely rare to find qP values quoted for plant-based systems. Compared to other expression systems, one of the limitations of plant cells is the fact that they are significantly larger than other cell types, such as mammalian and bacterial cells [2]. Due to the large size of plant cells, it is difficult to enhance productivity in suspension cultures by increasing cell numbers. Although protein yield in plant systems can be improved by increased production volume, it would require scaling up of plant cell suspension cultures, which is expensive and commercially not viable [63]. This is in direct contrast to whole plants that can produce high biomass at a low cost. Another interesting plant system that has been successfully used in pharmagenes expression and production is the hairy roots system with its more genetic stability to produce recombinant proteins for many years [64,65,66,67].

## 4. Transcriptional and Translational Challenges of Using Transgenic Plant Vectors and Strategies to Overcome Them

Post-transcriptional modifications are a challenge in recombinant protein production in plants. For this reason, many complex proteins are often produced in CHO cells because these cells allow genuine post-translational modifications such as glycosylation [2]. One way of eliminating transcriptional and translational problems during recombinant protein production is the use of plant-cell-free lysates. Lysates prepared from wheat germ embryos have been widely used because they possess everything that is needed for both transcription and translation. Washing during the preparation of extracts may eliminate translational inhibitors, allowing transcription-translation reactions to achieve yields as high as 100 μg/mL in a single batch process. The limitation of using wheat germ extracts is the fact that they are costly and require some time to prepare [68]. Not long ago, a cell-free lysate based on tobacco BY-2 cells that involves a coupled transcription-translation reaction in an 18-h batch process was shown to result in yields of up to 270 μg/mL [69].

## 5. Plant-Based Vaccine Production over the Past 5 Years (2017–2021)

### 5.1. Cancer Vaccines

Very recently, the tobacco plant was used in the production of the colorectal carcinoma-associated protein GA733-2. GA733-2 is a candidate protein in the production of a vaccine against colorectal cancer. The authors of [70] sought to improve the production rate of the GA733-2 protein, which has been previously produced in low quantities, something that limits its large-scale utilization. They used different combinations of tobacco species *Nicotiana benthamiana* and *Nicotiana tabacum* and the following three *Agrobacterium tumefaciens* strains: C58C1, LBA4404, and GV3101 to transiently express human colorectal carcinoma antigen GA733-2. The combination of *Agrobacterium tumefaciens* strain LBA4404 and the tobacco strain *Nicotiana benthamiana* resulted in the greatest yield of the recombinant proteins GA733-2 and GA733-Fc, with the highest expression level of recombinant GA733-2 recorded as 15.92 μg/g. After in vitro assay analyses, the tobacco-derived rGA733-2 and rGA733-Fc proteins were stable and bioactive.

Yiemchavee and co-workers [71] produced by transient expression in *Nicotiana benthamiana*, a chimeric protein made up of the extracellular domain of CTLA-4 (LTB-CTLA4) and the *E. coli* enterotoxin B subunit as a carrier. The resultant recombinant protein yield was up to 1.29 μg/g leaves FW after 4 days of infiltration. In mice, the plant-made LTB-CTLA4 induced humoral responses against both the CTLA-4 and the LTB moieties, an indication of its high immunogenicity.

The human papillomavirus (HPV), at an advanced stage, can cause cervical cancer, hence the need for its alleviation. Yanez and colleagues [72] investigated the potential plant expression of LALF_32–51_-E7, an HPV-16 candidate vaccine produced through the fusion of a modified E7 protein to a bacterial cell-penetrating peptide (LALF). They employed different expression vectors and found that the greatest yield of LALF_32–51_-E7 was achieved using a self-replicating plant expression vector and chloroplast targeting compared to cytoplasm localization. This system allowed the large-scale production of LALF_32–51_-E7, a feature that would eliminate costs attached to mammalian vaccine production as plant expression systems are very cost-effective. In a follow-up study, in order to ascertain the association between LALF_32–51_-E7 and the chloroplasts, Yanez and co-authors [73] fused the LALF_32–51_-E7 gene with one encoding enhanced GFP to produce an LG fusion protein. Transmission electron microscopy confirmed the localization in the chloroplasts wherein small structures that look like protein bodies (PBs) were observed.

C-C Chemokine Ligand 21 (CCL21) functions as an anticancer protein through its co-localization with T cells and dendritic cells in tumors to bring about antimetastatic immunity. A study by [74] attempted to express the CCL21 recombinant protein in a tomato plant (*Solanum lycopersicum*) through agroinfiltration. CCL21 was synthesized and cloned into pBI121. The consequent CCL21 plasmid was then agroinfiltrated into tomato leaves. After 3 days, recombinant CCL21 protein was expressed, with greater expression observed in the transformed leaves. The scratch assay showed the role of this protein in antimetastatic properties.

The plant recombinant protein prod”ctio’ system has been shown to be an effective tool in the production of monoclonal antibodies. Ofatumumab, a monoclonal anti-CD20 antibody, was expressed in *Nicotiana benthamiana* [75]. Ofatumumab is an anti-cancer antibody. Jin and others [75] wanted to make a comparison between the affinities of ofatumumab produced in transgenic plants and that produced in CHO cells. They developed ofatumumab with and without an HDEL tag. There was higher expression, in plants, of ofatumumab that was tagged with HDEL than in the untagged one. There was also a significant reduction in the binding affinities of both the tagged and untagged plant-derived ofatumumab compared to the CHO cell-derived ofatumumab. The complement-dependent cell cytotoxicity efficacy of CHO-cell-derived ofatumumab was significantly higher compared to that of both the HDEL-tagged and untagged plant-derived ofatumumab [75]. These results show that ofatumumab derived from plants, due to its poor affinity, may not be a perfect example of monoclonal body production in plants; however, it may help in the study and understanding of post-translational modifications.

The human epidermal growth factor 2 (HER2) is a principal marker of breast cancer development. Therefore, antibodies to HER2 are requisite for alleviating breast cancer. However, the production of these antibodies is very costly. Recently, anti-HER2 single-chain fragment variable (ScFv-Fc) was produced in *Arabidopsis thaliana* seeds using the promoter *Phaseolus vulgaris* β-phaseolin [76] (Dong et al., 2017). Recombinant anti-HER2 ScFv-Fc was successfully expressed in *Arabidopsis thaliana* seeds. In mature seeds, protein yield was as high as 1.1% of total soluble protein. Recently, anti-HER2 single-chain fragment variable (ScFv-Fc) was produced in *Arabidopsis thaliana* seeds using the promoter *Phaseolus vulgaris* β-phaseolin [77]. Recombinant anti-HER2 ScFv-Fc was successfully expressed in *Arabidopsis thaliana* seeds. In mature seeds, protein yield was as high as 1.1% of total soluble protein. The anti-HER2 antibody produced in *Arabidopsis thaliana* seeds had anti-tumor activity against the human breast cancer cell line SK-BR-3, suggesting the potential of the *Arabidopsis thaliana* seed protein system to express commercial antibodies.

In addition to cost, one of the challenges of recombinant protein production is that the expressed protein may not be identical to the native protein. The use of full-length proteins as antigens is a cost-effective and convenient strategy. The use of full-length proteins as antigens means that antibodies against multiple epitopes across the sequence will be generated. However, since the resulting antibodies are against multiple epitopes, it is highly possible that these antibodies may recognize other proteins with homologous epitopes, leading to non-specific cross-reactivity [78] (Liew et al., 2021). Although non-specific antibodies can be eliminated by affinity purification of serum, it does not eliminate cross-reactivity against homologous epitopes contained within the full-length protein sequence. In any case, affinity purification is not possible for all proteins due to solubility differences.

Immunization with a corresponding peptide of a full-length protein allows antibody development without the full-length protein but only the amino acid sequence. Because peptide sequences are relatively short, they offer epitope diversity and better specificity than full-length proteins [78]. To compensate for the likely inability of peptides to elicit immune responses because of their low molecular weight, they are often coupled with carrier proteins prior to immunization. This helps to enhance the immune responses and leads to the generation of antibodies against both the carrier proteins and the peptide sequences. However, serum affinity purification is only performed against the peptide sequences so that only antibodies that bind to the peptide sequences are isolated. This is a huge advantage because the isolated antibodies, after purification, can recognize multiple epitopes on the peptide sequence, offering better antibody affinity than monoclonal antibodies, which recognize single epitopes [78].

Edible vaccines provide a system that allows the consumption of genetically engineered plants with genes that specify an antigen that triggers a mucosal immune response. Because plant-made antigens are encapsulated in plant tissues, this protects them from the highly acidic conditions of the stomach. A lot of research has been directed toward polio vaccines over the years. Some of the disadvantages of polio vaccines are that oral polio vaccines (OPVs) contain attenuated viruses that have the potential to revert to their pathogenic forms, while the intestinal mucosal immunity provided by inactivated polio vaccines (IPVs) is quite insufficient. Additionally, generally, these vaccines are expensive and inaccessible to poor communities and are highmaintenance as their stability must be preserved. In the continued effort to address such shortcomings, recently, polio viral proteins (VPs) 1 and 2 were expressed in carrot cell lines [79]. Oral immunization of mice resulted in elevated levels of S-IgA and IgG as detected in the fecal excrement of the mice. This may be a cost-effective and safer approach to the induction of sustainable mucosal immunity against polio. Because hepatitis B DNA-based immunization is costly and requires unique storage requirements, attention is shifting toward the development of oral vaccines in plants. Recently, the HBsAG gene was successfully transformed into a tomato plant (*Solanum lycopersicum* L.) [80], indicating progress in using tomatoes as a viable production system for edible vaccines.

### 5.2. HIV Vaccines

The human immunodeficiency virus (HIV) can cause acquired immunodeficiency syndrome (AIDS) if it is not suppressed. The current use of antiretroviral drugs to suppress HIV has been effective in reducing AIDS-related mortality. However, antiretroviral drugs are expensive and mostly inaccessible to people in poorer countries, who are the ones in dire need of these drugs. Therefore, PMP offers a cost-effective platform to produce HIV vaccines in plants [5]. Although up to today, exactly four decades since the first reported case of HIV, a commercialized HIV vaccine has yet to be realized. Research continues to be performed on developing one, including using transgenic plants as a vaccine production platform.

Broadly neutralizing antibodies (bNAbs) have been actively investigated in HIV research. The manufacture of a novel bispecific fusion protein, which comprises the antigen-binding fragment (Fab) of the CD4 binding site-specific bNAb VRC01 and the HIV-1 envelope glycan shield targeting antiviral lectin Avaren (VRC01_Fab_-Avaren), has been recently undertaken [81]. This fusion protein was expressed in *Nicotiana benthamiana* with the aid of the GENEWARE^®^ tobacco mosaic virus vector. After purification, VRC01_Fab_-Avaren displayed superior neutralizing activity than did the individual parent molecules VRC01 IgG and Avaren-Fc. IC_50_ values ranged between 48 and 310 pM [81], showing the ability of plant-based expression systems to provide a suitable environment for the production of bispecific anti-HIV proteins.

The manufacturing of soluble HIV Env gp140 antigens using a transient expression system in *Nicotiana benthamiana* was recently undertaken based on the virus isolates CAP256 SU and Du151 [82]. Immunization of rabbits with the lectin affinity-purified antigens resulted in increased titers of binding antibodies such as Tier 1 virus-neutralizing antibodies and antibodies against the V1V2 loop region. The removal of aggregated Env species by gel filtration led to the induction of better binding and neutralizing antibodies [82].

Although PMP provides an excellent platform for recombinant protein production from transgenic plants, for some proteins, post-translational modification remains a challenge. Recently, a suite of human chaperones were co-expressed in order to improve HIV-1 soluble gp140 vaccine production in *Nicotiana benthamiana* [83]. During the co-expression of calreticulin (CRT), there was an increase in the expression of Env as well as the amelioration of the endoplasmic reticulum stress response genes. A combination of the co-expression of CRT with the transient expression of human furin was undertaken to allow the manufacturing of an appropriately cleaved HIV gp140 antigen. Transient co-expression in plants allows the production of appropriately cleaved proteins at high yields.

Microbicides have been reported to block the entry of HIV into human cells. PMP and the use of transgenic plants in the production of recombinant proteins can be applied to produce microbicides. Griffithsin is an antiviral lectin that can inhibit HIV entry into cells at high potency. [84] used dried tobacco leaves to produce griffithsin. Griffithsin accumulated in stably transformed tobacco chloroplasts. The yield of griffithsin was as high as 5% of the total soluble protein of the plant. The plant-produced griffithsin was simple to purify and its capability to neutralize HIV was similar to that of griffithsin expressed in bacteria [84].

### 5.3. Vaccines for Other Diseases

The production of a hepatitis B vaccine continues to be attempted, mainly through the investigation of the hepatitis B core antigen (NBcAg). The expression of this antigen in yeast, *E. coli,* and other expression systems is well documented [85] Recently, this antigen was produced in the tobacco plant [86]. HBcAg was expressed at 110–250 mg/g FW in stably transformed *Nicotiana tabacum*. This, consequently, shows that tobacco, due to its massive leaf biomass, can be used to produce HBcAg at output levels that are not dissimilar to those observed when using other recombinant protein transient expression systems. After purification with sucrose, HBcAg retained its antigenicity and its CLP structure. The intramascular introduction of 2 × 10 μg of the purified HBcAg antigen in mice resulted in a significant response evidenced by a serum anti-HBc titer of around 150,000 [86]. This response was statistically equivalent to that induced by the reference antigen. There was an increase in the anti-HBc IgG isotypes IgG1 and IgG2a during the immune response. There was also an induction of IgG2b and IgG3 in mice innoculated with the plant-derived antigen [86]. These findings point to the potential of tobacco-derived HBcAGg antigen as a vaccine candidate against chronic hepatitis B.

Dengue fever remains a global health challenge. The Sanofi Pasteur-produced Dengvaxia vaccine was recently authorized for use against the dengue virus. Dengvaxia is a live attenuated tetravalent dengue vaccine consisting of four dengue virus types encoding chimeras of the structural envelope (E) and pre-membrane (prM) genes in combination with the yellow fever 17D vaccine strain nonstructural gene. These four chimeric dengue vaccine candidates were first cultured in Vero cells, following which they were combined to make a single vaccine formulation.

However, Dengvaxia offers no immunity to children below the age of nine. A humanized dengue vaccine was recently developed. Using plant expression, poly-immunoglobulin G scaffold (PIGS) was fused to the consensus dengue envelope protein III domain (cEDIII) to produce an IgG Fc receptor-targeted vaccine candidate [87]. In transgenic mice expressing human FcγRI/CD64, the vaccine-induced neutralizing antibodies as well as cell-mediated immunity indicated its immunogenicity. Furthermore, the purified polymeric fraction of dengue PIGS (D-PIGS) led to more intense immune activation when compared to the monomeric form, indicating better interaction with the low-affinity Fcγ. These findings may signal the potential of plant-expressed D-PIGS in dengue vaccine manufacturing.

Synucleinopathies are a group of neurodegenerative diseases such as Parkinson’s disease and dementia. Currently, there is no cure for these diseases. Recently, [1] produced LTB-Syn in carrot cell lines to produce oral vaccines that do not require purification. LTB is a chimeric antigen produced in plants that comprises the subunit B of the enterotoxin from enterotoxigenic *E. coli* and three B cell epitopes from α-Syn. The LTB-Syn yield was as high as 2.3 μg/g of dry biomass [1]. There was high stability of the antigen encapsulated in lyophilized carrot cells. The LTB-Syn was able, in mice, to prime immune responses that induced significant humoral responses. This makes the carrot-made oral LTB-Syn vaccine a candidate that deserves further investigation.

Multiple sclerosis, an autoimmune disorder characterized by inflammatory demyelination, remains a medical challenge. Three peptides (BV13S1, BV5S2, and BV6S5) have been identified as multiple sclerosis vaccine candidates. Arevalo-Villalobos and colleagues [1] recently produced, using tobacco plants, the three peptides simultaneously by developing a polypeptide that comprised the sequences of the peptides. They designed the polypeptide with a picornaviral 2A sequence in between so that the individual peptides could be released after translation. The levels of accumulation of the BV13S1, BV5S2, and BV6S5 individual peptides were as high as 0.5 μg/g FW leaves. In mice, oral immunization of the tobacco-made peptides induced humoral responses.

Dental caries or cavities, commonly called tooth decay, are mainly caused by bacterial infection, especially *Streptococcus mutans*, of the oral cavity. Therefore, any anti-tooth decay activity should involve developing immunity against infections by the causative bacteria. Recently, Bai and others [88] developed a fusion anti-caries DNA vaccine (PAcA-ctxB) via the fusion of the A region of the cell surface protein PAc (PAcA) coding gene of *Streptococcus mutans* with the cholera toxin B subunit coding gene (CTB). Using agrobacterium-mediated plant transformation, these plasmids were then integrated into the tomato genome. The presence of transgenes in the tomato genome was confirmed both at the transcript and protein levels, an indication that transgenic tomatoes are potentially viable for the development of antigens against human caries.

Although poliomyelitis has been put under control globally, current polio vaccines have shortcomings. Some are very expensive, while others are made from attenuated viruses, which have the potential to become pathogenic. These shortcomings can be remedied using transgenic plants. Recently, Bolaños-Martínez and colleagues [89] investigated the potential expression of polio antigens (VP1, VP2,VP3, and VP4) in the tobacco species *Nicotiana tabacum*. The expression of these VPs in tobacco cells was confirmed. The oral and subcutaneous administration of these VPs induced systematic humoral and local responses, signaling their potential candidacy in the production of cost-effective and safe polio vaccines.

The human papillomavirus (HPV), which is closely associated with breast cancer, cannot be cultured in vitro. Therefore, virus-like particles are used for HPV vaccine production. HPV vaccines are very expensive and therefore emphasize the need for alternative HPV vaccine production systems, such as the use of transgenic plants. L1, the major structural protein of the HPV-16 capsid protein, was expressed in tobacco (*Nicotiana tabacum)* chloroplasts [90]. In mature plants, there was a high yield of about 3 mg L1/g FW, the highest HPV L1 level of expression observed in plants. The recombinant L1 protein aggregated into virus-like particles and showed conformation-specific epitopes. The intraperitoneal injection of L1 transgenic plant leaf protein extracts in mice resulted in high immunogenicity.

Table 3 below summarises some of the vaccines that have been expressed in plants over the past five years (2017–2021).

## 6. Adjuvant as a Vaccine Delivery System Tried for Plant-Based Immunogens

Adjuvants are added during vaccine formulation to improve the immunogenicity of antigens. Most adjuvants are molecules or chemicals acquired from infectious agents or from plant proteins that have immunomodulatory capacities [94]. In recent times, because of their pharmaceutical potential, some plant molecules such as polysaccharides, saponins, and lectins have been under exploration as adjuvant formulation candidates [95]. Principal considerations in the development of plant-derived immune stimulatory compounds such as adjuvants is their ability to boost immune responses while remaining non-toxic. These compounds should be applicable to different vaccine formulations, such as mucosal vaccines. Based on their effector mechanisms, there are the following three types of adjuvants: types A, B, and C. Type A adjuvants such as monophosphoryl lipid A are pattern recognition receptor (PRR) agonists [96], while type B adjuvants such as alum hydroxide interact with antigen-presenting cells (APCs) and antigens [97]. Type C adjuvants such as CD28 super agonist antibodies interact with co-stimulatory molecules on APCs.

Limitations of vaccines such as inadequate induction of immune response have motivated increased vaccine delivery systems that can improve the immunogenicity and stability of antigens. Edible and intradermal vaccine formulations have been shown to induce both mucosal and systemic immune responses. On the other hand, injectable vaccines require specialized storage and transport conditions. They have been reported to induce robust systemic humoral responses and result in weak T cell-mediated mucosal protection and immunity [98]. As a result, current plant-vaccine research activities are aimed at alternative vaccine delivery methods. For example, some studies have investigated the intradermal delivery of vaccines or their delivery to the mucosal interface to allow the quick and wide distribution of the antigen in the body and the ability to induce both protective mucosal and systemic humoral and cellular responses. In contrast to injectable vaccines, several advantages of intradermal and edible vaccines have been reported. These include reduced cost, improved antigen stability, cold chain elimination, as well as the fact that these vaccines are self-administered [99].

## 7. Optimization of Recombinant Vaccine Recovery and Purification

### 7.1. Downstream Processing of Recombinant Proteins

Upstream processing is the first part of the biopharmaceutical production process involving the growth of the transgenic plant or plant cell cultures or plants transiently expressing a given biopharmaceutical and concerns the preparation of growth media, plant transformation protocols under controlled conditions to express and manufacture a specific biopharmaceutical product, and extraction of the raw material required for the production of the candidate product. Downstream processing refers to the purification, recovery, and concentration of a given biopharmaceutical from the complex bulk plant material. This may include formulation steps signifying the transformation from the drug molecule to the drug product. This could be performed at the scale of the laboratory, pilot, or bulk manufacturing, focusing on process optimization, manufacture scale-up, and troubleshooting. Furthermore, this may involve the management of the plant’s resources and biohazards. Available systems for biopharmaceutical manufacturing vary based on the plant type and the site of localization of the protein product (chloroplast, apoplast, cytoplasm, or vacuole). Which option to choose depends on the essential characteristics necessary for the final product. For instance, mAbs are typically secreted into the apoplastic space via the endoplasmic reticulum/golgi secretory system and subsequently recovered from extracts of plant material through standard filtration as well as chromatographic approaches. In other instances, the product biopharmaceutical could be retained within the plant tissue, such as encapsulation in the chloroplasts with the requirement of minimal processing of the plant biomass yielding a feed additive to be formulated for oral delivery. Plant molecular pharming has provided a platform for the cost-effective and large-scale production of recombinant proteins in transgenic plants. However, for plant-based recombinant protein production systems to be fully utilized, the challenges encountered during downstream processing must be addressed. The downstream processing of recombinant proteins expressed in transgenic plants is disrupted by the presence of high levels of secondary metabolites such as alkaloids in phenolics, which are present in crude extracts of some plants such as tobacco [100]. However, some plants, such as lettuce, are known to produce very little of these secondary metabolites [16]. While a lot of attention has been placed on the upstream processing of recombinant proteins, such as employing specific vectors, promoters and expression cassettes, and other elements [101,102,103,104], more needs to be performed to address downstream processing, which relates to protein extraction and purification.

Most of the plant-based recombinant protein expression systems are faced with challenges of downstream processing because they are based on the intracellular expression of proteins in whole plants, which requires the disruption of the plant tissue to extract the product [105]. Downstream processing is a costly exercise since plant extracts possess a large portion of proteins from the host cell, which has to be removed [92].

Downstream processing activities such as the aqueous two-phase system, centrifugation, filtration (to remove debris), and flocculation have been shown to improve recombinant protein recovery and purification. Even though protein tagging can also facilitate the downstream processing of recombinant proteins, it is difficult to apply at a commercial scale and this has led to the preference of large-scale applicable strategies such as heat/pH precipitation and membrane technologies for the separation of host and target proteins in the cell [106]. Currently, efforts are being made toward the development of plant cell suspension cultures because they would allow the secretion of recombinant products directly into the culture media. This will circumvent the need for plant tissue disruption that is necessary to extract recombinant proteins from the plant cell. It is very difficult to limit whole plant-based protein expression systems within a clean room environment, something that is incompatible with good manufacturing practice (GMP) [107]. However, the use of contained systems in plant culturing has alleviated regulatory and safety reservations regarding the open-field production systems. The major disadvantage of contained systems is that they counteract some of the benefits of open-field production systems, such as cost reduction during upstream scaling up [105]. One of the earlier challenges in the downstream processing of recombinant proteins was that, because plant extracts are dissimilar in their composition, they require dissimilar conditioning strategies before they can be purified.

### 7.2. Downstream Processing of Plant-Derived Vaccine Candidates

Heat precipitation, or blanching, can remove about 80% of host cell proteins during the downstream processing of recombinant proteins, provided that the target protein has thermostability. Recently, Menzel and company [92] utilized both blanching and chromatography during the purification of FQS, a malaria vaccine candidate they had transiently expressed in the tobacco species, *Nicotiana benthamiana*. There was a threshold temperature of about 75 °C where the malaria monoclonal antibody 4B7 was unable to recognize the vaccine candidate, FQS, if this temperature was exceeded. A reduction in temperature from 80 °C to 75 °C restored antibody binding and the continual precipitation of the host cell proteins. Possibly because of the inactivation of proteases due to high temperature, heat precipitation blocked the degradation of FQS in the plant extracts, and chromatography revealed a 60% recovery of FQS and a final purity of approximately 72%.

An anti-Zika virus subunit vaccine, ZIKV E, was transiently expressed in *Nicotiana benthamiana* leaves and found to induce an immune response in mice [93]. Purification of the vaccine candidate using one-step Ni^2+^ affinity chromatography resulted in up to 90% homogeneity. One-step Ni^2+^ affinity chromatography involves the subjection of plant extracts to Ni^2+^-based immobilized metal anion chromatography (IMAC). Sucrose density gradient centrifugation has been used in the extraction and purification of a tobacco-made hepatitis B antigen, resulting in a 43% purification of the antigen [86].

The processes involved in the extraction and purification of recombinant proteins made in mammalian and microbial cells have been widely reported [59]. Unlike in other expression systems such as CHO cells, where products are secreted into the culture medium by the cells growing in suspension, and they are easy to recover and purify, recombinant proteins produced in plants must be extracted from the plant material. The purification of recombinant proteins produced by plant cell suspension culture is complicated by the fact that the medium contains multiple proteins secreted by the host cell [2] (Schillberg et al., 2019). Tobacco is one of the most used plant species in recombinant protein production because it results in a high biomass yield. However, protein storage in tobacco leaves is not very stable and its possession of toxic alkaloids complicates downstream processing, especially purification [59].

## 8. Conclusions

There is a lot of effort directed at the use of plants as model host species during recombinant protein production. Depending on the type of protein being produced and the production scale, the choice of a host plant is one of the most important considerations during the development of recombinant proteins in transgenic plants. The choice of a host is governed by factors such as the size of the product as well as post-translational modifications. Similarly, the transformation method chosen plays an equally central role during the recombinant protein production process. While attention has been placed on the upstream processing of plants during the early stages of recombinant protein production, more attention and research need to be directed at downstream processing. Different recombinant proteins require different extraction and purification processes, emphasizing the need for more extensive research in the recovery and purification of the proteins. The use of recombinant protein extraction and purification strategies such as blanching [92], sucrose density gradient centrifugation [86], and one-step Ni^2+^ affinity chromatography [93] has been used successfully during the downstream processing of plant-produced vaccine candidates. Despite current challenges and obstacles, the production of vaccine candidates in transgenic plants appears to be getting more pervasive.

Existing plant expression platforms afford a lot of benefits beyond the conventional advantages of high scalability, low cost, eukaryotic protein modification, and enhanced safety. Inimitable transient expression vectors have been generated to enable the expression of therapeutics and vaccines at an unprecedented pace to combat probable pandemics and bioterrorism threats. Plant-derived biologics comprise the fastest and largest growing category of pharmaceutical products. Presently, a majority of human biopharmaceuticals are generated in microbial and mammalian cell cultures. These require facilities that are capital-intensive in addition to the necessity of fermenters, costly downstream processing, cold temperature requirements for storage, and transport as well as sterile delivery schemes [108]. On the other hand, plant-based production platforms dispense with the need for bioreactors, expensive cell culture media, and capital-restrictive facilities while being easily scaled up in comparatively inexpensive greenhouses with facile mineral solutions. Therefore, plant-derived expression systems afford the innate advantage of reduced manufacturing costs. Studies by [109] showed that plant-based schemes require costs of materials being as low as $1–2 for every kg of the plant-derived protein product.

The profitability and market acceptability of any product are greatly impacted by manufacturing costs and have been a contentious topic due to the lack of ready availability of the actual costs of generating plant-based pharmaceuticals on an industrial scale. Nevertheless, the expenses associated with the downstream processing of plant-made biopharmaceuticals, particularly for parenteral administrations, have been estimated to be analogous to those of other protein expression systems.

Novel plant expression systems also provide speed and tractability that cannot be matched by expression technologies derived from mammalian cell cultures due to the modernization of expression vector development, especially vectors enabling transient expression. The advancement of ‘deconstructed’ viral vector systems such as pEAQ, magnICON, and geminviral expression platforms has successfully addressed issues such as inadequate protein expression levels, speed, and consistency of biopharmaceutical production in plants [108,110,111] For instance, by means of transient expression using these deconstructed viral vectors, expression of as high as 5 mg of mAb per gram of fresh leaf weight has been reported within 2 weeks, while, on the other hand, an equivalent production using transgenic plants can require several months to years. Additionally, transgenic plant platforms frequently result in inconsistent and reduced protein yields [112]. The high pace and level of production enabled by transient plant expression systems facilitate pre-clinical studies requiring milligram and gram levels of biopharmaceuticals. Moreover, ‘bridge’ versions of such vectors have been developed to scale-up biologic manufacturing in stable transgenic plants [113,114] Therefore, ‘deconstructed’ viral vectors afford versatile tools for rapid evaluation of candidate biologics towards transitioning them into commercial manufacturing on a large scale.

Plant systems also serve as novel vehicles for the oral administration of biopharmaceuticals. Traditional biologics are generated by expensive downstream processing and require a ‘cold chain’ for storage and transportation, in addition to sterile needles for parenteral administration. In this context, oral delivery of drug candidates is desirable. However, this has been elusive because of the denaturation as well as degradation of these products in the human digestive system, in addition to the inability of these drugs to traverse across the gut epithelium and deliver the drugs to target cells. However, the digestive enzymes present in the human gut cannot hydrolyze the carbohydrate glycosidic bonds in the plant cell wall, and therefore, plant cells can shield the expressed biopharmaceuticals from enzymes and acids in the stomach through bioencapsulation, thus allowing them entry into the gut lumen where they are released enzymatically by commensal bacteria in the gut [115]. Recent investigations show that orally delivered protein drugs encapsulated in plant cells can traverse the gut epithelium and enter the bloodstream. Then, based on the specific targeting sequences that they are fused to, these orally delivered biologics can elicit tolerance against the production of inhibitory antibodies associated with the injection of these molecules or gain entry into the circulatory system of the human host [115,116,117]. Plant-cell encapsulated protein drugs have been found to stably maintain their pharmacological efficiency even several years following storage at room temperature [76].

Such remarkable results suggest that biologics encapsulated in plant cells could represent a room-temperature-stable drug candidate that can be orally administered to patients and could be of great benefit to healthcare programs in resource-poor countries facing logistical challenges of parenteral delivery of effectual medicines. Furthermore, even in developed regions, this edible vaccination technology can decrease the costs involved in downstream processing and cold chain requirements for storage and transport.

## Figures and Tables

**Figure 1 vaccines-10-01861-f001:**
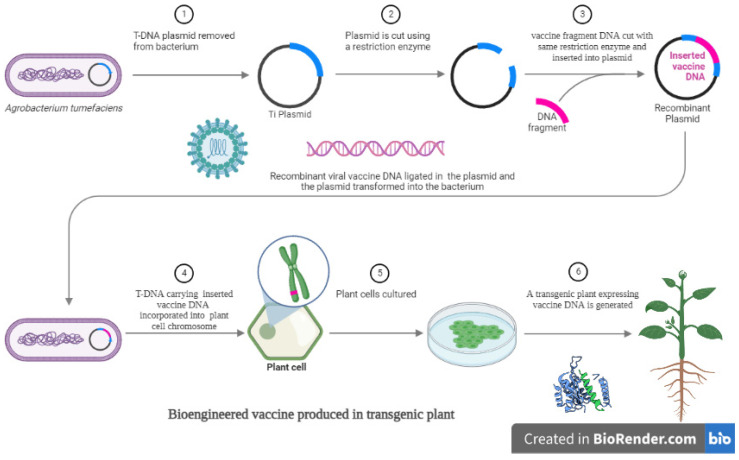
Bioengineered vaccine expressed and produced in transgenic plant.

**Table 1 vaccines-10-01861-t001:** Applications of different plant species in recombinant protein production.

Host Plant Species	Applications
Tobacco	- Production of cancer vaccines [17] and cancer antibodies [18,19] - Production of HIV vaccines [20,21]) and HIV antibodies [22]
Arabidopsis thaliana	- Expression of α creatine kinase MAK33 mAb (Fab) with up to 6% protein accumulation [23] - Production of the Helicobacter pylori TonB protein for immunization against Helicobacter infections [24].
Rice (Oryza sativa)	- Expression of recombinant hGM-CSF with high yield of about 129 mg/L [25] - Production of recombinant human lysozyme protein [26]
Maize (Zea Mays)	- Expression of β-glucuronidase (GUS) with accumulation levels of up to 0.7% of water-soluble protein extracted from dry seeds - Production of the protease inhibitor aprotinin [27]
Potato (Solanum tuberosum)	- Manufacturing of pharmaceutical proteins such as human interleukins [28,29] and human interferons [30] - Production of vaccines against different enteric diseases [31]
Tomato (Lycopersicon esculentum)	- Expression of different pharmaceutical proteins such as the scFv recognizing carcinoembryonic antigen [32]. - Expression of vaccines such as the respiratory syncytial virus-F protein [33]
Alga (Chlamydomonas reinhardtii)	- Production of monoclonal antibodies [15]
Lettuce (Lactuca sativa)	- Production of recombinant proteins such as the hepatitis B surface antigen [34,35] - Production of antibodies against colorectal cancer, rabies, and anthrax [36].

**Table 2 vaccines-10-01861-t002:** Advantages and disadvantages of different expression systems used during recombinant protein production (Developed with guidance by [59,60,61].

Expression System	Advantages	Disadvantages
Plant	- Cost effective - Quick to grow - Cheap to grow - Capacity to produce complex proteins - Growth protocols are optimized - No pathogenic contamination risks	- Low capacity for glycosylation - Regulatory approval lacking - Inconsistent product quality - Difficult downstream purification
Mammalian cells	- Protein yield is high - Funding by industry prevalent - Correct post-translational modifications - Regulatory approval exists - Proper protein folding	- Production process costly - Takes time to culture - Not easy to scale up - Human pathogen contamination risks - Expensive to culture - Heterologous product
Insect cells	- Capacity to produce complex proteins - High protein expression - Correct post-translational modifications - Proper protein folding - Regulatory approval exists - Easy to scale up	- Expensive to culture - Long production time - Unwanted post-translational modifications
Bacteria	- Regulatory approval exists - High protein expression - Easy to scale up - Manipulatable - Cell lines well characterized - Short production time - Cost effective	- No post-translational modifications - Accumulation of endotoxins - Improper folding of large proteins
Yeast	- Easy to scale up - Quick to grow - Cost effective - Regulatory approval exists - Proper protein folding - Manipulatable - Cell lines well characterized - Correct post-translational modifications	- Protein hyperglycosylation - Thick cell walls make cell disruption difficult - Low capacity for glycosylation

**Table 3 vaccines-10-01861-t003:** Some plant-made vaccine candidates produced over the past 5 years (2017–2021).

Recombinant Protein	Expression Plant Species	Disease	Method of Transformation	Level of Protein Expression	Reference
GA733-2	Tobacco *(Nicotiana benthamiana)*	Colorectal cancer	*Agrobacterium* *(Transient expression)*	15.92 μg/g	[70]
LTB-CTLA4)	Tobacco *(Nicotiana benthamiana)*	Cancer	*Agrobacterium* *(Transient expression)*	1.29 μg/g FW	[71]
LALF_32–51_-E7	Tobacco *(Nicotiana benthamiana)*	Human papillomavirus (HPV)	*Agrobacterium* *(Transient expression)*	0.017% TSP	[72]
Griffithsin	Tobacco *(Nicotiana tabacum)*	Human immunodeficiency virus (HIV)	Biolistic bombardment (Stable expression/Chloroplast)	5% TSP	[84]
NBcAg	Tobacco *(Nicotiana tabacum)*	Hepatitis B virus (HBV)	*Agrobacterium* (Stable expression/Chloroplast)	110–250 mg/g FW	[86]
D-PIGS	Tobacco *(Nicotiana benthamiana)*	Dengue virus	*Agrobacterium* (Transient expression)	17 mg/kg FW	[87]
LTB-Syn	Carrot (Daucus carota)	Synucleinopathies	*Agrobacterium* (Transient expression)	2.3 μg/g dry biomass	[91]
BV Proteins	Tobacco *(Nicotiana tabacum)*	Multiple sclerosis	*Agrobacterium* (Stable expression)	0.5 μg/g	[1]
L1	Tobacco *(Nicotiana tabacum)*	HPV	Biolistic bombardment (Stable expression/Chloroplast)	3 mg/g FW	[90]
FQS	Tobacco *(Nicotiana benthamiana)*	Malaria	*Agrobacterium* (Transient expression)	51 mg/kg	[92]
ZIKV E	Tobacco *(Nicotiana benthamiana)*	Zika virus	*Agrobacterium* (Transient expression)	160 μg/g FW	[93]

## Data Availability

Not applicable.

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
