# Peer review of "Recent Progress on Vaccines Produced in Transgenic Plants"

_vaccines, 2022, doi:10.3390/vaccines10111861_

Round 1

Reviewer 1 Report

Vaccines, Review: Recent progress on vaccines produced in transgenic plants, by G. Gaobotse, et al.

General comments:

The review needs to focus on approach as well as summarizing what has been done. This would include a section and in best situation a cartoon amplifying the various procedures employed in transforming plant cells to create transgenic cells.  The range of vectors and uniqueness of promoters needs to be discussed.  One assumes that direct DNA is used in particle bombardment, but only a guess.  The distinctions in methodology for getting the vector into chloroplasts, vs. nucleus, or merely cytoplasm needs explained in a cartoon.  Just when, and when not, is Agrobacterium use preferred? 

The section “Host plant species used …” is very reference intensive and hard to read.  A table documenting progress in the eight plant cell types would make the review much more accessible, accompanied by a discussion of why each cell type offers advantages and outlining the difficulties. The format used in Table 1 and section “Different expression systems…” would be most adequate.

The concept distinctions of expressing single epitopes, or combined strings of single epitopes vs. whole protein expression was not discussed.  Related to this, there is confusion about expressing and purifying a protein for use as immunogen to elicit an immune response, vs expressing protein complexes, such as various antibody subunits for use as reagents. 

The idea of expressing immunogens in plant cells (like the carrot or tomato examples) that can be consumed and used to induce mucosal immune response is a tremendous idea that was not fully developed nor distinguished from just purifying an expressed immunogenic protein for use as vaccine candidate. 

The concept of adjuvants, and vaccine delivery methods tried for the plant-produced immunogens was neglected, or not mentioned.

Plant folks have been touting the advantages of making vaccines in plant cells for several decades, but key statements as to why this has not caught on are treated lightly.  What I took as a lesson is that many plants, except lettuce produce secondary metabolites as phenolics and alkaloids that are hard to remove during purification of the expressed fusion proteins.  Endotoxins from bacteria can be eliminated (somewhat) easily.  Is this also possible for the alkaloids?  Are there common assays for them, or are they too diverse to test for?

We need a scale factor to consider the wide range in “recombinant protein yield” per plant (or other) expression systems.  More discussion is needed regarding why this varies widely.

There is a need to discuss the transcriptional and translational problems to be overcome in using plant cell transgenic vectors.

Improvable manuscript grammatical details:

L 30: should be “conventional” “c” is missing

L 48-49: No mention of viral vectors.  Are these also used with plant cell transformation?

L 58: remove word “different”

L 72: the concept of “agroinfiltration” may be common to plant people, but needs better explanation for general audience

L 78: Same as above comment for the concept of “transgenic plants”

Ls 80-115: How does chloroplast transformation differ from plant cell transformation.  Are the mitochondria also transformed (is there an effect)?

L 114: The term “idiotype vaccines” is broad and confusing.  Give more focus.

L129-130: basic terminology not fully clear. 

L143: suggest deleting comma after kinase to improve readability

L141: What does “easy transformation” mean?

Ls443-444: not fully clear what is meant

L450 and other places: what is kg/ha?  Should this be Kg/ha?

Ls453-456: Confusing sentence.  Perhaps start: Perhaps the biggest shortcoming of rice …

L527: Define “particle bombardment” and explain what is being bombarded and how done and the practical problems involved

L585: delete “in” before “30”

L622: Used many times, the term “recombinant protein” conveys many ideas.  Really what you are talking about is vector “expressed fusion protein”

L686: replace “grown” with excpressed

L689: replace “easy” with ease

Ls706-707: The concept of “uniform mammalian post translational modifications” is thrown out without any explanation

L712: some mechanism (conjugation mech?) and range of action for Agrobacterium being able to  “introduce foreign DNA” into plant cells is needed

 L776 …section “Plant-based vaccine …:  Many instances of using the reference format as a noun.  For example, in line 790 state: “Yiemchavee, et al. (2021) produced by…”

Ls780-781:  Something is missing or deleted.  Can’t make sense of what is implied.

L803: the word “the” should be substituted with a more specific word (e.g., current, present, mammalian)

L807: suggest adding semicolon after word “chloroplasts” and eliminate “as”

Ls819-826:  Is there some reason for underlining all the section?

L826:  Remove “Additionally” and  start with “The complement-”…

Ls828-831: Sentence starting with “Even” is confusing.  Revise.

L840 and several other spots:  use of “a signal” in place of “suggesting” is confusing and misleads reader

L 849: Period after “realized”  Research continues…

L853: change :manufacturing” to manufacture

L853: delete the second “of”

L856-857: “This not “the” fusion protein…

L857: delete “then”

L858: delete “its”

L859: insert “did” between than the

L860: delete “which showed” and replace with showing

L867: use “virus” not viruses

L871: change to “modification remains”

L874: remove comma

L880:  delete “There are”  start with “Microbicides”… name some and give reference(s) 

L882: If don’t make above change, then delete “these”

L884:  Delete “There was accumulation” to “Griffithsin accumulated…”

L886-887: Revise sentence

L891: “is well documented” (Needs at least one reference added here”

L893:  What does xxmg/gFW” mean?  I may have missed the definition but somewhere it should be spelled out

L905: What expression system was used to produce Denvaxia vaccine?

L1017-1018: Terms “upstream” and “downstream” processing need to be defined

L1033-1035: The concept of “limit whole plant-based expression” needs more explanation

L1039: concept of “open-field production systems” requires more explanation

L1042: delete “the fact”

Author Response

We have addresses all the questions, comments and suggested changes as you can see in the attached file

Reviewer 2 Report

The authors present an interesting review on the state of vaccine production in transgenic plants.  I disagree however with the authors' characterization of expression systems other than plants.  The authors argue that plants are more cost effective than other expression methods without really giving adequate justification.  Much of the cost of producing a GMP protein will come from its purification in a GMP setting which would be similar for the different expression systems.  The high protein yields of some of the other expression systems simplifies this process, e.g. grams per liter for some mammalian expressed antibodies, and reduces the overall cost.  Time is also an issue and the turn around time for bacterial, yeast or even mammalian systems can be much quicker than that of a plant such as tobacco.  Although in my opinion the review's presentation is upside down from the current state of affairs, the authors do highlight some interesting examples such as those few cases for edible vaccines where purification would not be an issue. 

Minor points:  line 30 conventional is missing a c, line 445 improbable should be problematic, line 454 should be is that it is not just is, line 628 It should be They, line 630 It should be They, and lines 818-825 should not be underlined unless the authors explain why.

Author Response

We answered all the questions and suggestions made by reviewer 2 as you can see in the attached file

Round 2

Reviewer 1 Report

Thanks to the authors for substantial revision.  Good!

Re the inserted cartoon: 

The legend has word "vaccives" should be "vaccines".  Also, the middle part of cartoon shows "reinserted into bacterium".  Is this an integration event, or merely "transformation" of the recombinant plasmid into bacterium.  This is confusing and makes a difference, since you are dealing with Ti plasmid that has site specific recombinases and it is unclear if you imply they are involved.

Otherwise, thanks for making these improvements to your original draft.

Author Response

Answers to reviewers

Re the inserted cartoon: 

The legend has word "vaccives" should be "vaccines".  Also, the middle part of cartoon shows "reinserted into bacterium".  Is this an integration event, or merely "transformation" of the recombinant plasmid into bacterium.  This is confusing and makes a difference, since you are dealing with Ti plasmid that has site specific recombinases and it is unclear if you imply they are involved.

We corrected the cartoon accordingly as shown in the correct version

Otherwise, thanks for making these improvements to your original draft.

Thanks

Reviewer 2 Report

The current version of the manuscript appears to be a more balanced representation of the field.

Author Response

The current version of the manuscript appears to be a more balanced representation of the field.

Thanks